# Sliding Mode Controller with Generalized Extended State Observer for Single Link Flexible Manipulator

Tushar Bhaskarwar [1], Huzein Fahmi Hawari [2,*], Nursyarizal B. M. Nor [2], Rajan Hari Chile [1], Dhammaratna Waghmare [3] and Sumit Aole [1]

1. Department of Instrumentation Engineering, Shri Guru Gobind Singhji Institute of Engineering and Technology, Nanded 431606, India; bhaskar.tushar90@gmail.com or 2017pin604@sggs.ac.in (T.B.); rhchile@yahoo.com (R.H.C.); sumit.aole@gmail.com (S.A.)
2. Department of Electrical & Electronic Engineering, Universiti Teknologi PETRONAS, Seri Iskandar 32610, Malaysia; nursyarizal_mnor@utp.edu.my
3. Model Based Design, Department of Research and Development in Punch Powertrain, Pune 410506, India; dhammasunny22@gmail.com
* Correspondence: huzeinfahmi.hawari@utp.edu.my

**Abstract:** This paper presents an enhanced generalized extended state observer (EGESO) based sliding mode control (SMC) technique for dealing with the disturbance attenuation problem for a class of non-integral chain systems with mismatched uncertainty. In the proposed control law, the robust SMC with reaching phase elimination is applied in the proposed control law, which uses the estimated states of a system. The stability analysis is thoroughly examined for both EGESO and SMC. The efficacy of the proposed controller is verified using specific examples, and later it is applied on a single-link flexible manipulator. Through simulation and experimentation analysis, it is observed that the proposed controller is giving a robust transient response as compared to existing GESO based controllers.

**Keywords:** generalized extended state observer (GESO); mismatched system; sliding mode controller; non-integral chain form; real time experimentation





## 1. Introduction

One of the eminent problems in modern control theory is the precise modeling of systems for robust control design. The second major challenge in robust control design is tackling unmodeled internal nonlinearities or unmeasurable external disturbances that may reduce the efficiency of the closed-loop system [1–4]. Implementing disturbance estimation techniques, which can eliminate these uncertainties and modeling errors, is always helpful in controller design due to the increase in the importance of precise control. Various disturbance observer-based control (DOBC) techniques [5–11] have been developed in the past few years. The survey paper [10] will provide a complete overview of the DOBC study. As per [10], disturbance observer (DOB) [5], unknown input observer (UIO) [11], and extended state observer (ESO) [8] are the most widely researched and applied among these disturbance estimation approaches.

Although many researchers have utilized DOBC on various systems, satisfying matching conditions, very few studies addressed systems with mismatched uncertainties [12–15]. ESO based control (ESOBC) only requires the relative order of system and approximate model information [16–18]. Originally ESOBC design was in the form of a typical chain of integrators with a matching condition considered [19], which restricts the ESOBC applicability to the mismatched system with non-integral chain form of systems [16]. Recent studies [13,14] suggested the use of GESO for such systems. The above survey motivates us to augment states observer quality of EGESO [14] with robust sliding mode controller to improve the performance of systems with mismatched uncertainty. This technique

reduces the number of sensors required for plant states to be measured and makes the plant cost-effective.

An SMC is one of the famous, robust, nonlinear control techniques for uncertain systems because of its simple design procedure [20]. SMC was previously used for aircraft, robotic manipulators, batteries, and photo-voltaic systems [21], marine vehicles, and process control applications [22,23]. The designing of sliding mode control requires a sliding function to reach the desired behavior of a sliding surface and a discontinuous controller that pushes the system trajectories towards the sliding (switching) manifold in a finite time and stays there for the future [24–26]. An SMC is sensitive to lumped uncertainty before it reaches the sliding surface, so it is necessary to eliminate this reaching phase to make the system insensitive to uncertainties [12,27].

The ESO-based SMC [28–32] approach has been utilized previously but lacked in addressing issues like chattering in SMC, reaching phase elimination, non-integral chain form, mismatched system, and proper disturbance estimation. To address some of these issues, many researchers [33–36] used a single-link flexible manipulator (SLFM) model as an application, but it fails to address all the issues and to achieve the desired performance. Due to the intrinsic underactuated nature of SLFM, designing a controller for such systems is a challenging task.

In this paper, the EGESO based SMC, along with reaching phase elimination, is proposed for SLFM along with the mismatched uncertainties and succeeded in reducing the tip chattering ($\alpha$) of the flexible arm in real-time experimentation. The proposed controller is found to be effective in set-point tracking, handling mismatched uncertainties, and reducing the chattering effect in comparison to [13,14,30].

The remainder of the paper is structured as follows. Section 2 reviews the requisite preliminaries for problem statement identification. Section 3 defines the proposed control law along with the design of an EGESO and a modified stable sliding surface, with the elimination of reaching phase property, and EGESO stability is also indicated here. The proposed closed-loop system control law's stability is described in Section 4. The model description and dynamic equations with system matrices are shown in Section 5. Section 6 demonstrates the reliability of the proposed controller with numerical simulation and experimental results and a comparison of the performance indices. Then the conclusion ends the paper.

## 2. Preliminaries and Problem Statement

A class non-integral chain form of an uncertain non-linear system with $n^{th}$ order and mismatched disturbances is considered in [13] as:

$$\begin{cases} \dot{x} = Ax + B_u u + B_f d(x, w(t), t), \\ y = Cx, \end{cases} \tag{1}$$

where $x = [x_1 \cdots x_n]^T$, $u \in R$, $y \in R$, and $w \in R$ are the state vector, input, controlled output, and external disturbances respectively. $A \in R^{n \times n}$, $B_u \in R^{n \times 1}$, $B_f \in R^{n \times 1}$, $d(\cdot) \in R$ may be mismatched uncertain disturbance function of $x$ and $w$ and $C \in R^{(1 \times n)}$.

**Remark 1.** *The generalized term for lumped disturbance $d(x, w(t), t)$ in Equation (1), is assumed to be bounded. This uncertainty function $d(x, w(t), t)$ includes unmodeled dynamics, external noise, and parametric uncertainty, which may be challenging for a simple feedback controller.*

**Remark 2.** *The matching condition with respect to disturbance matrix $B_f$ is given by $B_u = \lambda B_f$, $\lambda \in R$ [13]. Equation (1) describes a more specific class of systems because the system (1) is not only restricted to an integral chain form and it can also be subject to mismatched uncertainties [37].*

The GESO for system (1) with state feedback controller and disturbance compensation gain law was proposed in [13], whereas the enhanced GESO based controller for the same

class of system is given in [14].

The control law provided by [13] is stated as,

$$u = K_x \hat{x} + K_f \hat{d} \tag{2}$$

where

$$K_f = -[C(A + B_u K_x)^{-1} B_u]^{-1} C(A + B_u K_x)^{-1} B_f \tag{3}$$

disturbance compensation gain is denoted as $K_f$, and state feedback gain is represented as $K_x$. $\hat{x}$ and $\hat{d}$, are the estimations of $x$, $d$, respectively, and are obtained by following GESO structure:

$$\begin{bmatrix} \dot{\hat{x}} \\ \dot{\hat{x}}_{(n+1)} \end{bmatrix} = \begin{bmatrix} A_{(n \times n)} & B_f \\ 0_{(1 \times n)} & 0 \end{bmatrix} \begin{bmatrix} \dot{\hat{x}} \\ \dot{\hat{x}}_{(n+1)} \end{bmatrix} + \begin{bmatrix} B_u \\ 0 \end{bmatrix} u + L_0(y - [C\ 0_{1 \times 1}]\hat{x}) \tag{4}$$

where $\hat{x} = \begin{bmatrix} \hat{x}_1 & \hat{x}_2 & \cdots & \hat{x}_n \end{bmatrix}^T$ are the estimates of the state variable $\begin{bmatrix} x_1 & x_2 & \cdots & x_n \end{bmatrix}^T$, $L_0$ is observer gain matrix with dimension $(n + 1) \times 1$, which has to be design, and $\hat{x}_{(n+1)}$ is an estimation of lumped disturbance $d$. The GESO simultaneously provides an estimate of the states and the uncertainties, as the uncertainties are part of the states of the extended order system [13].

**Remark 3.** *The lumped disturbances assumed to be bounded and they must have constant steady state value as $t \rightarrow \infty$ i.e., $lim_{t \rightarrow \infty}\ \dot{d}((x, w(t), t)) = 0$, means $lim_{t \rightarrow \infty}\ d((x, w(t), t)) = C_d$, where $C_d$ is a constant value. Under this assumption and control structure given from (2) to (4), the EGESO based control gives stable bounded response.*
*Here Remark 3 is completed, whereas the enhanced generalized observer-based control law proposed by [14] stipulates as,*

$$u = K_x \hat{x}_g + u_g \left( \hat{\eta}, \hat{d}, \dot{\hat{d}}, \cdots, \hat{d}^{(k_d)} \right) \tag{5}$$

*where $K_x$ is selected so that $\bar{A} \cong A + B_u K_x$ will be Hurwitz, $\hat{x}_g$ is estimated EGESO states which consist of estimation of original states along with extended state and it is given as $[\hat{x}\ \hat{d}\ \dot{\hat{d}}\ \cdots\ \hat{d}^{(k_d)}]^T$, $u_g$ is a function of combined term which consists of $\hat{\eta}$, $\hat{d}$, $\dot{\hat{d}}$, and $\hat{d}^{(k_d)}$ which are the dynamic component, estimation of lumped disturbance, estimation of disturbance derivative, and estimation of disturbance $k_d^{th}$ order derivative respectively. According to [14], $\hat{\eta}$ is estimated using Equation (6) and it is given as,*

$$\left.\begin{array}{l} \dot{\zeta} = \phi\zeta + \Gamma\hat{d}, \\ \hat{\eta} = H\zeta \end{array}\right\} \tag{6}$$

*where $\phi \in R^{P \times P}$ (Hurwitz), $\Gamma \in R^P$, $H \in R^{1 \times P}$, $p = m_{u_m}^{tot}$, and the detailed calculation of $u_g$ function is discussed in [14]. This additional $u_g$ function in Equation (5) helps to reject disturbances much more effectively as compared to Equation (2). The above given preliminaries are related to paper [13,14], based on which the problem statement has been defined, and the newly proposed control law is designed.*

In this paper sliding mode, controller is introduced instead of state feedback controller, with EGESO to strengthen transient response of a system. The proposed scheme is shown in Figure 1 and control law is indicated by dotted box. The limitation of constant steady state disturbance for control law (2) is eliminated by proposed EGESO based sliding mode controller. The minimum values of performance indices obtained through proposed control law indicate its superiority over (5). The estimated value of lumped disturbance is used in such a way that the effect of disturbances from output can be compensated without the need to know actual disturbance present in the system.

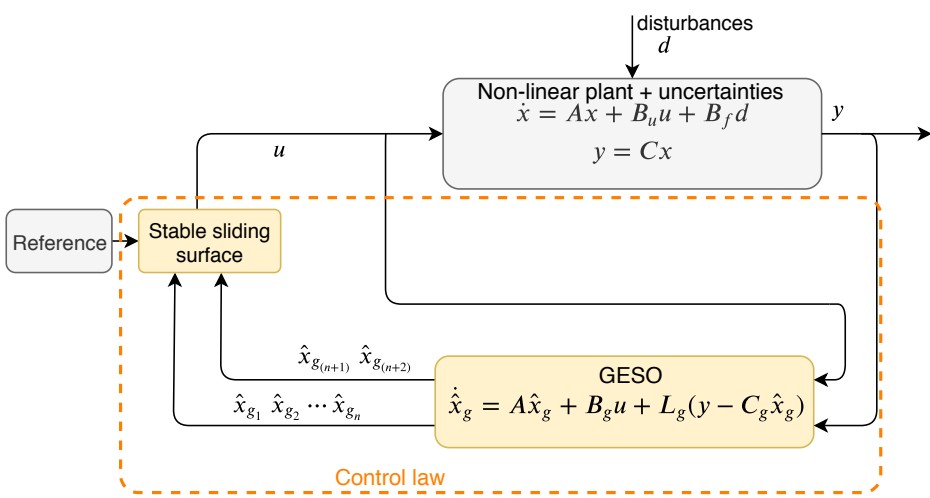

**Figure 1.** Block diagram of the proposed control scheme.

## 3. Proposed Control Scheme

A block diagram of the proposed control scheme has a structure given in Figure 1.

### 3.1. Criteria for EGESO

The appropriate criteria are made here just for designing extended state observer as per the paper proposed by [14]:

1. $\bar{A} \cong A + B_u K_x$, where $A$ and $B_u$ are any general system and control matrices respectively. See numerical examples in results section for more understanding.

2. $o_{u,j} \in \mathbb{C}$, $j = 1, \ldots, m_u$, with $o_{u,j} \neq o_{u,j+1}$, $\forall j$, denotes zeros in the matrix triple of $(\bar{A}, B_u, C)$, where $C$ is output matrix. For more information, see numerical examples in the results section.

3. $n_{u,j} \in \mathbb{N}$ denotes the multiplicity of $o_{u,j}$.

4. The zeros $o_{u,j}$, and their respective multiplicities $n_{u,j}$, are split into: minimum phase zero, $o_{u_m,j}$, $n_{u_m,j}$, $j = 1, \ldots, m_{u_m}$; non-minimum phase zero, $o_{u_{nm},j}$, $n_{u_{nm},j}$, $j = 1, \ldots, m_{u_{nm}}$; and zeros at the imaginary axis, $o_{u_0,j}$, $n_{u_0,j}$, $j = 1, \ldots, m_{u_0}$; satisfying $m_u = m_{u_m} + m_{u_{nm}} + m_{u_0}$.

5. The total number of zeros in the matrix triple $(\bar{A}, B_u, C)$ is indicated by $m_u^{tot} = \sum_{i=1}^{m_u} n_{u,i}$. Furthermore, in the same way: $m_{u_m}^{tot} = \sum_{i=1}^{m_{u_m}} n_{u_m,i}$, $m_{u_{nm}}^{tot} = \sum_{i=1}^{m_{u_{nm}}} n_{u_{nm},i}$, and $m_{u_0}^{tot} = \sum_{i=1}^{m_{u_0}} n_{u_0,i}$ are the total number of minimum phase zeros, non-minimum phase zeros, and zeros at the imaginary axis, respectively.

6. The matrix triple $(\bar{A}, B_f, C)$ uses the same notation as defined in (2)–(5), replacing the subindexes '$u$' by '$f$', where $B_f$ is a disturbance matrix. See numerical examples in results section for more understanding.

### 3.2. Design of Enhanced Generalized Extended State Observer

This sub-section is intended to design mismatched disturbance and states estimation method. The lumped disturbances shown by matrix $B_f$ are effectively estimated by (7). The EGESO is defined as

$$\left. \begin{array}{c} \dot{\hat{x}}_g = A\hat{x}_g + B_g u + L_g(y - C_g \hat{x}_g) \\ \hat{y} = C_g \hat{x}_g \end{array} \right\} \tag{7}$$

where $L_g \in R^{(n+2) \times 1}$ is the observer gain matrix which needs to be designed, and

$$A_g = \begin{bmatrix} A & B_f & 0_{n \times k_d} \\ 0_{k_d \times n} & 0_{k_d \times 1} & I_{k_d \times k_d} \\ 0_{1 \times n} & 0_{1 \times 1} & 0_{1 \times k_d} \end{bmatrix}, B_g = \begin{bmatrix} B_u \\ 0_{(k_d \times 1)} \\ 0_{1 \times 1} \end{bmatrix}, C_g = \begin{bmatrix} C \\ 0_{1 \times 1} \\ 0_{1 \times k_d} \end{bmatrix}^T, \tag{8}$$

$$x_g = \begin{bmatrix} x & d & \dot{d} & \cdots & d^{(k_d)} \end{bmatrix}^T.$$

are the generalized extended observer states, $A_g$ is new GESO system matrix, $B_g$ is GESO control matrix, $C_g$ is new GESO output matrix, $k_d = \max\{0, m_f^{tot} - m_u^{tot}\}$ and $\hat{x}, \hat{d}, \hat{\dot{d}}, \cdots, \hat{d}^{(k_d)}$ are estimations of $x, d, \dot{d}, \cdots, d^{(k_d)}$ respectively.

**Assumption 1.** *The matrix pair $(A, B_u)$ is controllable and the pair $(\bar{A}, C)$ is observable. Furthermore, the observability of matrix pair $(\bar{A}, C)$ is a necessary condition for the observability of $(A_g, C_g)$, as given in [13].*

**Assumption 2.** *The matrix $(\bar{A}, B_u, C)$ does not have zeros at the imaginary axis.*

**Assumption 3.** $d^{(k_d+1)}$ *is bounded.*

Here we define estimation error of the ESO as $e_0 = x_g - \hat{x}_g$ with $x_g$ defined in (8), by considering the observer dynamics as given in Equation (7). The following statement indicates throughout proof for boundedness of error $e_0$,

**Proposition 1.** *Under Assumptions 1–3, the boundedness of error $e_0$ is guaranteed if $L_g$ in (7) is chosen such that $(A_g - L_g C_g)$ is Hurwitz.*

**Proof.** By considering the system in Equation (1), the boundation of $e_0$ is given by the following proof.

$$\begin{cases} \dot{x} = Ax + B_u u + B_f x_{d,0} \\ x_{d,0} = d \\ \dot{x}_{d,0} = x_{d,1} = \dot{d} \\ \vdots \\ \dot{x}_{d,k_d-1} = x_{d,k_d} = d^{k_d} \\ \dot{x}_{d,k_d} = d^{(k_d+1)} \end{cases} \tag{9}$$

where $x_{d,k_d} = d^{k_d}$, value of $k_d$ varies from 0 to positive integer, and it is given as $k_d = \max\{0, m_f^{tot} - m_u^{tot}\}$. Disturbance terms can be expressed as $[x_{d,0}, x_{d,1}, \cdots, x_{d,k_d}]^T = [d, \dot{d}, \cdots, d^{k_d}]^T$. Using dynamics given in Equation (9), we are estimating disturbance and their derivatives from Equation (10). Expressing Equation (9) in matrix form leads to,

$$\dot{x}_g = A_g x_g + B_g u + B_{g,f} d^{k_d+1} \tag{10}$$

with $x_g, A_g, B_g$ defined in (8) and $B_{g,f} = [0_{1 \times (n+k_d)}, 1]^T$.

By differentiating $e_0 \triangleq x_g - \hat{x}_g$ and inserting Equations (7) and (10) in it, we get

$$\dot{e}_0 \triangleq (A_g - L_g C_g)(x_g - \hat{x}_g) + B_{g,f} d^{k_d+1} \tag{11}$$

which is bounded for any bounded $f^{k_d+1}$, since $(A_g - L_g C_g)$ is Hurwitz [38]. The ESO stability proof is finished here. □

### 3.3. Design of Stable Sliding Surface

This sub-section deals with designing a stable modified sliding surface $(\sigma_m)$ to eliminate mismatched disturbance of the system (1). Equation (23) shows that modified sigma's $(\sigma_m)$ dynamics do not depend on signum function, and hence, a considerable amount of chattering effect is removed from the traditional sliding mode controller. The estimated disturbance is used to design a stable sliding surface, and the control input is derived for tracking the system on the input trajectory.

A similar type of mismatched system as depicted in (1) is considered with second order form,

$$\begin{cases} \dot{x}_1 = x_2 + f(x) + e^{x_1} + w, \\ \dot{x}_2 = b(x) \cdot u, \\ y = x_1. \end{cases} \tag{12}$$

where $x = [x_1, x_2]^T$ are the system states, $b(x)$ is invertible and not zero, and $f(x)$ is the system disturbance. $d(x,t) = d = f(x) + e^{x_1} + w$ is considered as lumped disturbance. where external disturbance $w = 3$ acts on a system at time $t = 6$ s.

The stable sliding mode surface $\sigma$ is chosen as,

$$\sigma = c_1 x_1 + x_2 + \hat{d} \tag{13}$$

where $c_1$ is a user defined positive constant, $\hat{d}$ is the lumped disturbance estimation. By differentiating Equation (13) and incorporating (12) leads to

$$\dot{\sigma} = c_1 x_2 + c_1 d + b(x)u + \dot{\hat{d}} \tag{14}$$

and control $u$ is given by,

$$u = u_{eq} + u_n \tag{15}$$

putting Equation (15) in (14), Equation (14) modifies to

$$\dot{\sigma} = c_1 x_2 + c_1 d + b(x)u_n + b(x)u_{eq} + \dot{\hat{d}} \tag{16}$$

Equivalent control law, $u_{eq}$ is calculated from system nominal or known parameters. The remaining unknown terms have either lumped uncertainty or disturbance will be taken care of by $u_n$, and it is given by Equations (17) and (18) respectively,

$$u_{eq} = -b(x)^{-1}(c_1 x_2 + k_s \sigma + \dot{\hat{d}}) \tag{17}$$

where $k_s$ is positive constant chosen by designer.

$$u_n = -b(x)^{-1}(c_1 \hat{d}) \tag{18}$$

by substituting both (17) and (18) in (16), we get

$$\dot{\sigma} = \tilde{d} - k_s \sigma \tag{19}$$

where $\tilde{d}$ is disturbance estimation error and it is given by $\tilde{d} = d - \hat{d}$.

### 3.4. Control Law Using Estimated States

From (11), $x_1 = \hat{x}_1$, $x_2 = \hat{x}_2$, and so on till $x_n = \hat{x}_n$ and $d = \hat{x}_{(n+1)}$. A modified sliding surface using the estimated state is given as,

$$\sigma_m = c_1 \hat{x}_1 + \hat{x}_2 + \hat{d} \tag{20}$$

The control law is derived using similar steps from (14)–(18) on Equation (20),

$$u_{eq} = -b(x)^{-1}(c_1 \hat{x}_2 + k_s \sigma_m + \dot{\hat{d}}) \tag{21}$$

$$u_n = -b(x)^{-1}(c_1 \hat{d}) \tag{22}$$

After incorporating (21) and (22) in $\dot{\sigma}_m$ it leads to,

$$\dot{\sigma}_m = -k_s \sigma_m + \tilde{d} \tag{23}$$

where $\tilde{d}$ is disturbance estimation error and it is given by $\tilde{d} = d - \hat{d}$.

*3.5. Reaching Phase Elimination*

A study of reaching phase elimination is presented in [24]. Subsequently, due to the initial condition of states, it may be possible to get a high extensive control signal requirement and also to have insensitivity of lumped disturbances; reaching phase can be eliminated and it is designed as,

$$\overset{\star}{\sigma}_m = \sigma_m - g(x, t) = \sigma_m - \sigma_m(0)e^{-\alpha t} \tag{24}$$

where $g(x, t) = \sigma_m(0)e^{-\alpha t}$ and at time $t = 0$, $g(x, 0) = \sigma_m(0) \implies \overset{\star}{\sigma}_m(0) = 0$, whereas, at $t \to \infty$, $g(x, \infty) = 0 \implies \overset{\star}{\sigma}_m(\infty) = \sigma_m$.

The proposed control law given in (13) is modified as per reaching phase elimination rule (24) for studied examples. The importance of reaching phase elimination can be observed in the result section, in which LESOSMC controller results are compared with the proposed controller.

## 4. Stability

The stability of Equation (20) is given in this section. The EGESO stability is already presented in Equations (9)–(11), which proves that error dynamics of original and estimated states are going to zero, and it remains in bounds of $\mu$.

**Proposition 2.** *The ultimate boundedness of the sliding surface $\sigma_m$ is found by defining the following Lyapunov function.*

**Proof.**

$$V_s(\sigma_m) = \frac{1}{2}\sigma_m^2 \tag{25}$$

By differentiating Equation (25) it leads to,

$$\dot{V}_s(\sigma_m) = \sigma_m \dot{\sigma}_m \tag{26}$$

From the dynamics of $\sigma_m$ in Equation (23), we get

$$\begin{aligned} \dot{V}_s(\sigma_m) = &\ \sigma_m \left(-k_s \sigma_m + d - \hat{d}\right) \\ = &\ \sigma_m \left(-k_s \sigma_m + \tilde{d}\right) \end{aligned} \tag{27}$$

$$\dot{V}_s(\sigma_m) \leq -|\sigma_m|(k_s|\sigma_m| - |\tilde{d}|) \tag{28}$$

Using the bounds of $\tilde{d}_m$ found from GESO estimation, we obtain that

$$\dot{V}_s(\sigma_m) \leq -|\sigma_m|(k_s|\sigma_m| - \mu) \tag{29}$$

From Equation (29), for stability we must $K_s|\sigma_m| > |\mu|$. Thus, sliding surface is ultimately bounded by the bounds given by

$$|\sigma_m| \leq \frac{\mu}{k_s} \tag{30}$$

From Equation (30) it is noted that the bounds of $\sigma_m$ may have any value and it depends on the value of $k_s$ and therefore designer has to choose the value of $k_s$ as per performance requirement; this proves the asymptotic stability of an entire system. □

## 5. Simulation and Experimental Results

A comparative study has been carried out to show the efficacy of the proposed controller. In Example 1, EGESO based SMC is compared with other control techniques [13,14]. The proposed controller is also applied to Example 2, an uncertain, unstable system with minimum phase zero, and the results are compared with other techniques. Finally, the proposed controller is validated with other controllers on SLFM actuated by DC motor. Mismatched disturbances are introduced in all cases and removed actively from the output with the help of the proposed control law.

### 5.1. Study Example 1

A second order nonlinear system with mismatching condition is considered for study [13].

$$\begin{cases} \dot{x}_1 = x_2 + e^{x_1} + \omega(t) \\ \dot{x}_2 = -2x_1 - x_2 + u(t) \\ y = x_1 \end{cases} \tag{31}$$

being $d(x, \omega(t), t) \triangleq e^{x_1} + \omega(t)$ with $\omega(t) = 0$, $0 \leq t < 5$ s and $\omega(t) = 3$, $t \geq 5$ s. The system matrices are:

$$A = \begin{bmatrix} 0 & 1 \\ -2 & -1 \end{bmatrix}, B_u = \begin{bmatrix} 0 \\ 1 \end{bmatrix}, B_f = \begin{bmatrix} 1 \\ 0 \end{bmatrix}, C = \begin{bmatrix} 1 & 0 \end{bmatrix} \tag{32}$$

The modified sliding surface with reaching phase elimination as per (20) and (24) is considered as

$$\begin{aligned} \overset{\star}{\sigma}_m &= \sigma - \sigma(0)e^{-\alpha_s t} \\ &= c_1\hat{x}_1 + \hat{x}_2 + \hat{x}_3 - (c_1 + \hat{x}_3)e^{-\alpha_s t} \end{aligned} \tag{33}$$

where $\hat{x}_3$ is an estimation of disturbance $d$. Equivalent control law $u_{eq}$ and disturbance caring term $u_n$ is obtained as

$$u_{eq} = -[c_1\hat{x}_2 - 2\hat{x}_1 - \hat{x}_2 + \alpha_s c_1 e^{-\alpha_s t} - \dot{\hat{d}}\alpha_s e^{-\alpha_s t}] - k_s\overset{\star}{\sigma}_m - \dot{\hat{d}} \tag{34}$$

$$u_n = -[c_1 d + \hat{d}\alpha_s e^{-\alpha_s t}] \tag{35}$$

where the sliding surface coefficient $k_s = 5$, control gain $c_1 = 20$, observer poles kept at $s = -5$ (same, as placed in [14]), $\alpha_s = 4.5$, step size of simulation = 0.001 s, solver used in simulation is ode4 (Runge-Kutta), saturation limit for control signal considered as +30 to $-40$. It should be noted that the sliding surface coefficient $k_s$ should satisfy the Equation (30) which was generally considered at least 10 times greater than that of bounds $\mu$. Whereas, $c_1$ is dependent on the location of the pole to be placed in such a way that the sliding surface will always be stable. This system is a non-minimum phase and it is observed that the proposed controller gives better performance even after adding constant disturbance from $t \geq 5$ s as shown in Figure 2.

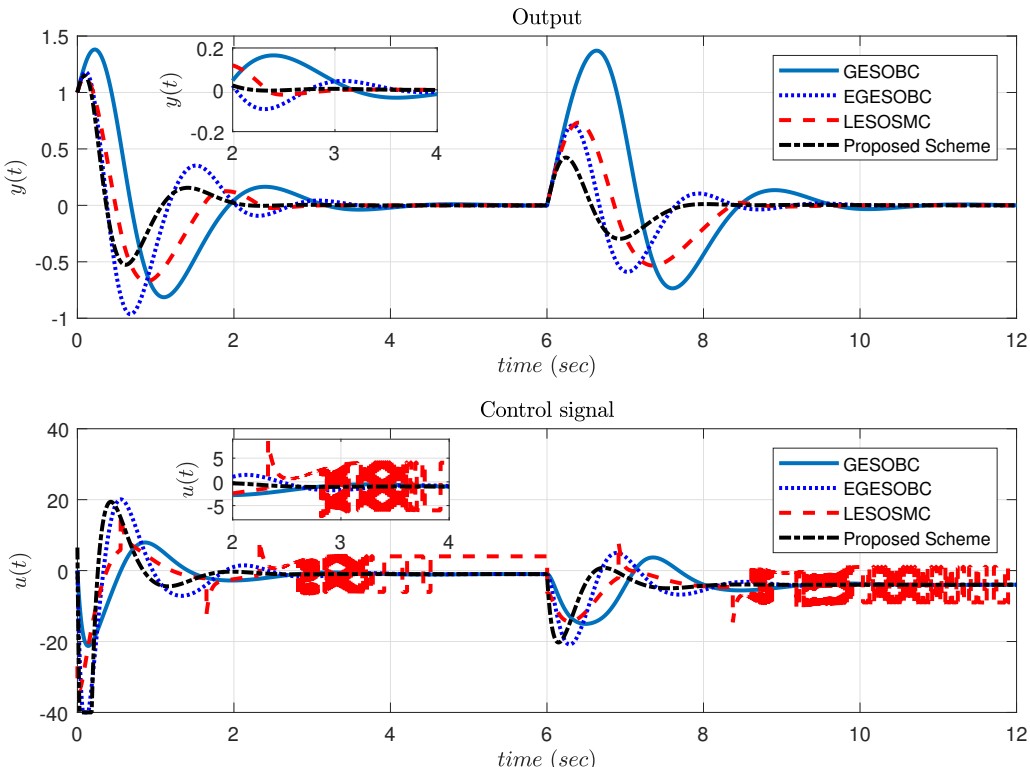

**Figure 2.** Response of output and control signal for Example 1.

If the control law is designed with $\sigma$ instead of $\overset{\star}{\sigma}_m$, then sliding surface starts initially from 20, whereas with reaching phase elimination it started from 0, which provides less control efforts at initial stage and better performance as shown in Figure 3.

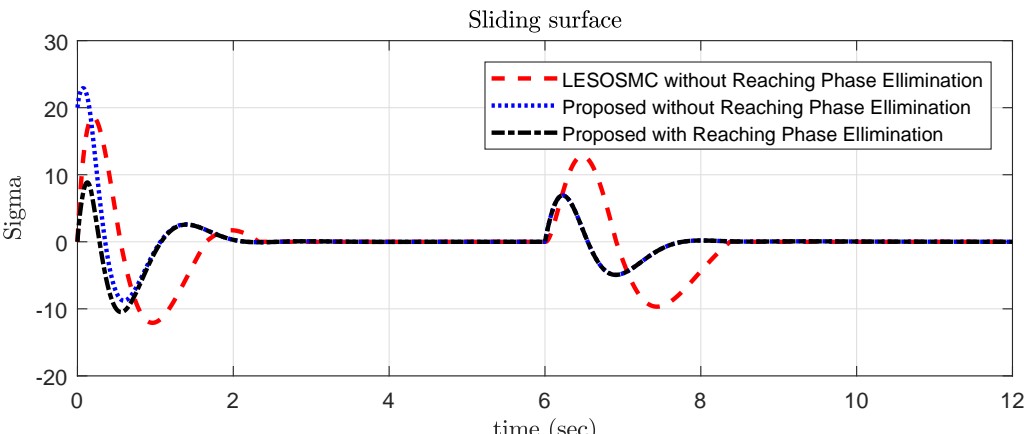

**Figure 3.** Sliding surface response for Example 1.

The error dynamics of actual and estimated states are given in Figure 4. The quick and perfect convergence of estimated states towards original states provides less error bound $\mu$. As this is an example of regulating type, the set point is zero. The performance indices for this system are shown in Table 1. From the performance indices, it is clarified that the proposed controller provides less error than other techniques in all cases.

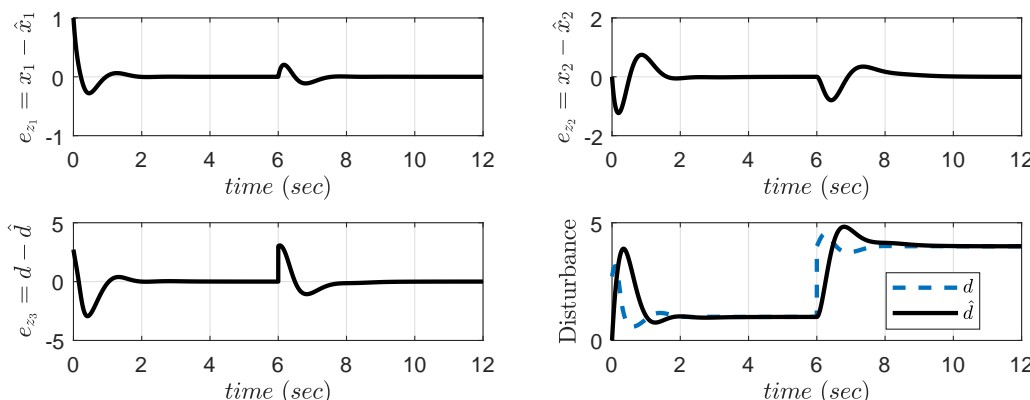

**Figure 4.** Error dynamics of proposed controller with reaching phase elimination for Example 1.

**Table 1.** Comparison of performance indices for Example 1.

| Controller | Set-Point ($r$) | Error Indices | | | | |
|---|---|---|---|---|---|---|
| | Output ($y$) | IAE ($10^{-2}$) | ISE ($10^{-2}$) | ITAE | ITSE ($10^{-2}$) | Control Efforts ($u$) in ITAE |
| GESOBC | error $= y - 0$ | 311.5 | 256.1 | 13.47 | 1030 | 280.5 |
| EGESOBC | error $= y - 0$ | 175.5 | 103.9 | 5.827 | 254.4 | 284.4 |
| LESOSMC | error $= y - 0$ | 179.4 | 101.8 | 6.910 | 322.3 | 355.5 |
| Proposed scheme | error $= y - 0$ | 96.66 | 47.43 | 2.736 | 71.81 | 259.4 |

### 5.2. Study Example 2

A third order uncertain unstable system having an internal minimum-phase zero is considered for study [14]:

$$\begin{cases} \dot{x}_1 = 3x_1 - 1.5x_2 + 0.5x_3 + 2u(t), \\ \dot{x}_2 = 2x_1, \\ \dot{x}_3 = x_2 + \tanh(x_3) + w(t) \\ y = 0.25x_2 + 0.75x_3 \end{cases} \tag{36}$$

being $d(x, w(t)) \triangleq \tanh(x_3) + w(t)$, with $w(t) = 0$, $0 \leq t < 5$ s and $w(t) = 5(t-5)e^{-\frac{(t-5)}{2}}$. The system matrices are:

$$A = \begin{bmatrix} 3 & -1.5 & 0.5 \\ 2 & 0 & 0 \\ 0 & 1 & 0 \end{bmatrix}, B_u = \begin{bmatrix} 2 \\ 0 \\ 0 \end{bmatrix}, B_f = \begin{bmatrix} 0 \\ 0 \\ 1 \end{bmatrix}, C = \begin{bmatrix} 0 & 0.25 & 0.75 \end{bmatrix}.$$

The proposed sliding surface designed with reaching phase elimination for this case is obtained as

$$\begin{aligned} \overset{\star}{\sigma}_m &= \sigma - \sigma(0)e^{-\alpha_s t} \\ &= c_1 y + \dot{y} - 0.75(d)e^{-\alpha_s t} \end{aligned} \tag{37}$$

where $d$ can be replaced by estimated state $\hat{x}_4$. Equivalent control law $u_{eq}$ and disturbance caring term $u_n$ is obtained as

$$u_{eq} = -[(0.25c_1 + 0.75) \cdot 2\hat{x}_2 + 0.75c_1\hat{x}_2 + 0.5(3\hat{x}_1 - 1.5\hat{x}_2 + 0.5\hat{x}_3)] - k_s\overset{\star}{\sigma}_m \tag{38}$$

$$u_n = -[0.75c_1\hat{x}_4 + 0.75\dot{\hat{x}}_4] \tag{39}$$

where the sliding surface coefficient is $k_s = 20$, control gain is $c_1 = 50$, observer poles are kept at $s = -5$ (same as placed in [14]), $\alpha_s = 2$, step size of simulation is 0.001 s, the solver

used in simulation is ode4 (Runge–Kutta), and saturation is not considered in control signal for this example. The efficacy of the proposed controller is better as compared to others as shown in Figure 5. The sliding surface response is shown in Figure 6 and it is observed that sliding surface starts exactly from 0 value.

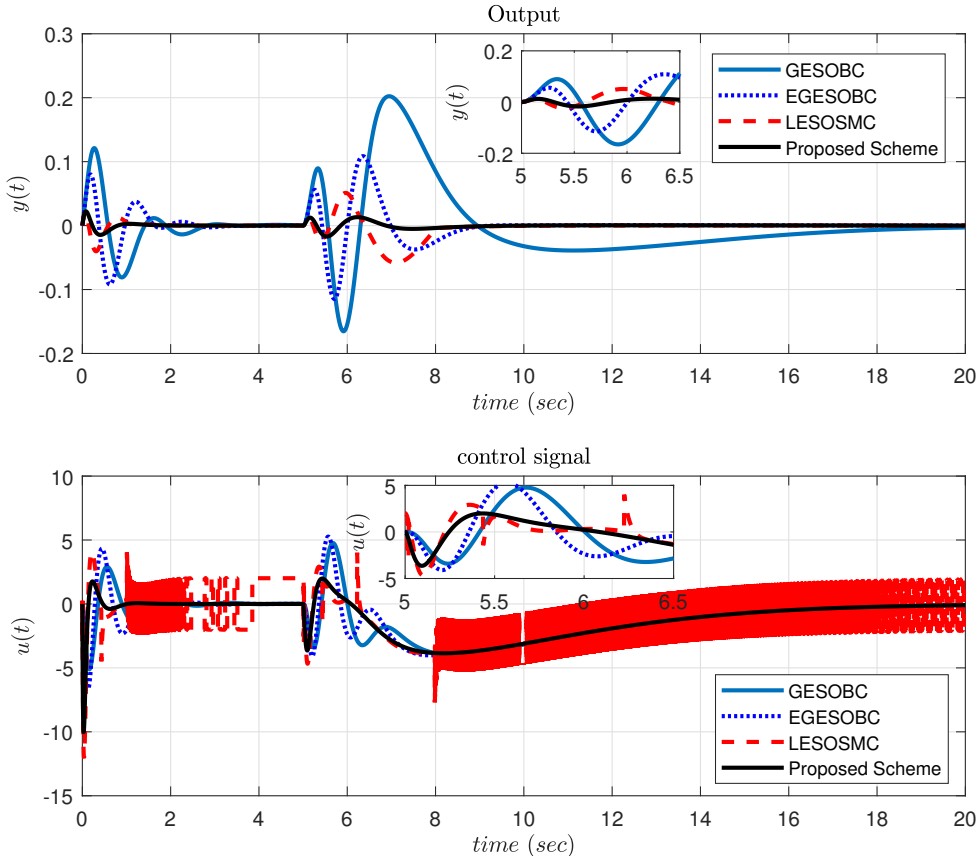

**Figure 5.** Response of output and control signal for Example 2.

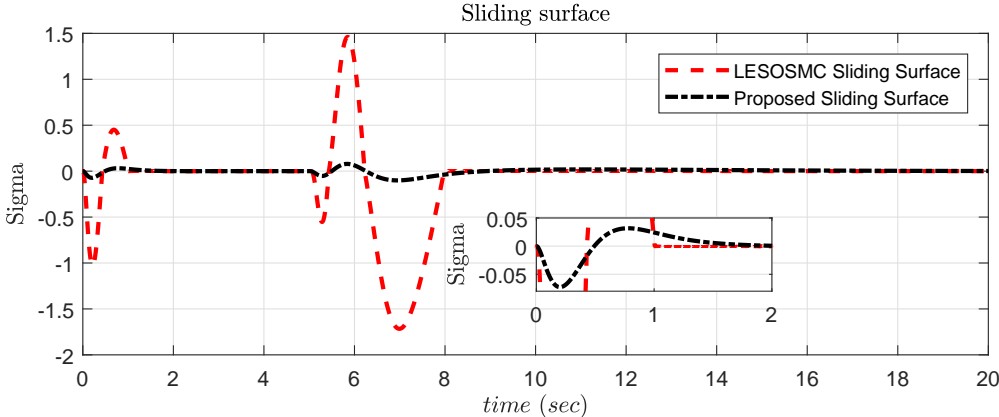

**Figure 6.** Sliding surface response for Example 2.

The error dynamics of actual and observed states are shown in Figure 7, and it is verified that it is approaching towards zero, whereas real disturbance and its estimation are also presented in Figure 7. The performance indices for this example are shown in Table 2. It is clear from the performance indices that less error is provided by the proposed controller than by other compared techniques in all aspects.

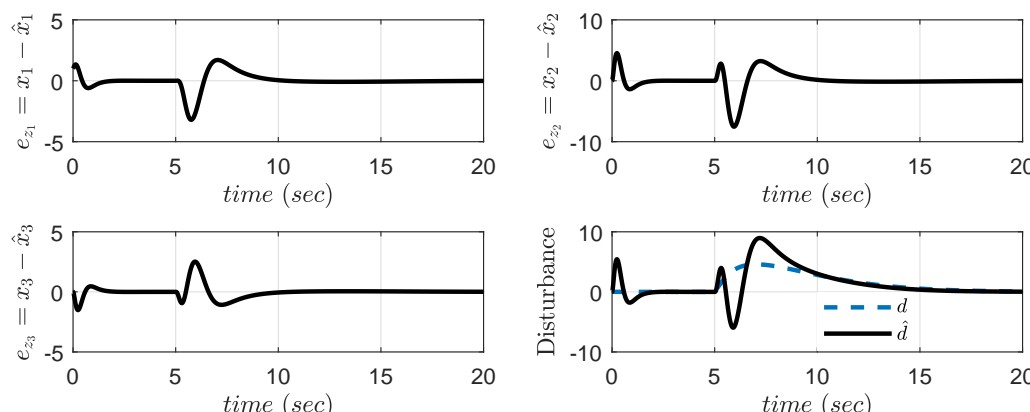

**Figure 7.** Error dynamics of proposed controller with reaching phase elimination for Example 2.

**Table 2.** Comparison of performance indices for Example 2.

| Controller | Set-Point ($r$) Output ($y$) | Error Indices | | | | Control Efforts ($u$) in ITAE |
|---|---|---|---|---|---|---|
| | | IAE ($10^{-3}$) | ISE ($10^{-3}$) | ITAE ($10^{-2}$) | ITSE ($10^{-3}$) | |
| GESOBC | error = y − 0 | 712.9 | 67.52 | 569.8 | 456.7 | 246.5 |
| EGESOBC | error = y − 0 | 233.3 | 14.50 | 113.7 | 67.89 | 248.2 |
| LESOSMC | error = y − 0 | 113.3 | 4.173 | 66.9 | 25.97 | 455.3 |
| Proposed scheme | error = y − 0 | 35.95 | 0.3193 | 19.60 | 1.314 | 238.1 |

### 5.3. Experimental Model

The elastic rotary attachment is a perfect test for modeling a dynamic relation to a robot or a spacecraft. The model emphasizes the effects on robot control systems of flexible link connections. Figure 8 shows a schematic diagram of the SLFM. Because of the flexible structure of the link, the end-point is shifted, as shown in the diagram, whenever it is actuated by an angle theta ($\theta$) at the servomotor end. Illustrations of various parameters are given in Table 3. The link's flexibility is modeled as a linear torsional motion.

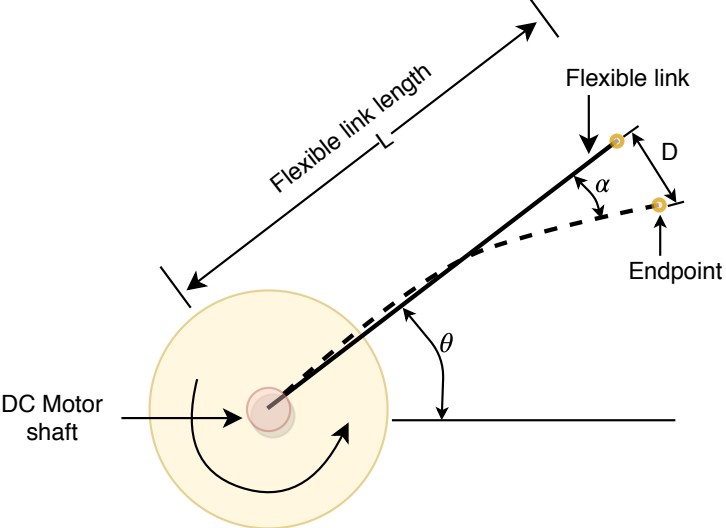

**Figure 8.** Flexible link manipulator schematic diagram.

**Table 3.** Nomenclature used for parameter.

| Parameter | Description | Unit |
|---|---|---|
| $\alpha$ | Arm end-point deflection | degree |
| $\theta$ | Servo motor gear angle | degree |
| $L$ | Flexible link length | cm |
| $T_l$ | Load torque | Nm |
| $D$ | Link end-point deflection (Arc length) | cm |
| $B_{eq}$ | Viscous damping coefficient | |
| $K_{stiff}$ | Total stiffness of model | Nm/deg |
| $J_{link}$ | Moment of inertia of link | Kg-m$^2$ |
| $J_{eq}$ | Equivalent moment of inertia of the model | Kg-m$^2$ |

### 5.3.1. System Model Dynamics

Dynamic equations of a system were derived using the formulation of Euler–Lagrange. Considering that the end-point deflection angle is minimal, this is estimated by $\alpha = \frac{D}{L}$, where $D$ is the link end-point displacement, and the length of the link is $L$. For modeling the flexible link, the 2nd order rotary system model is considered in this experimentation. The one end of the link turns around, and the other stays fixed. The simplified, flexible link manipulator model is obtained in [39], and by computing Lagrangian and later using the Euler–Lagrange equation, the flexible link manipulator dynamic equations are given by (40) and (41).

$$\ddot{\theta} = \frac{T_l}{J_{eq}} - \frac{B_{eq}\dot{\theta}}{J_{eq}} + \frac{K_{stiff}\alpha}{J_{eq}} \tag{40}$$

$$\ddot{\alpha} = -K_{stiff}\left(\frac{1}{J_{eq}} + \frac{1}{J_{link}}\right)\alpha + \frac{B_{eq}}{J_{eq}}\dot{\theta} - \frac{T_l}{J_{eq}} \tag{41}$$

Defining states as $x_1 = \theta$, $x_2 = \alpha$, $x_3 = \dot{\theta}$, $x_4 = \dot{\alpha}$, and $V_m = u$, the states representation of Equations (40) and (41) are described as per [33] given as (42).

$$\left.\begin{array}{l} \dot{x}_1 = x_3 \\ \dot{x}_2 = x_4 \\ \dot{x}_3 = 936.39x_2 - 41.19x_3 + 72.4593u \\ \dot{x}_4 = -1372.6x_2 + 41.19x_3 - 72.4593u \end{array}\right\} \tag{42}$$

The control problem is to synthesize EGESO based sliding mode law for controlling the servo motor deflection ($\theta$) as well as end-point deflection ($\alpha$), which is considered as a mismatched uncertainty in the first channel of (42). The experimental setup is shown in Figure 9.

### 5.3.2. Simulation Results for SLFM

The proposed EGESO based sliding mode controller is applied to 4th order uncertain system with matched uncertainty considered in 3rd channel as given in (42). Indeed, it is found that the disturbance does not affect the SLFM system in the same channel as the control action, which demands a complex closed-loop control [40]. State $x_3$ with matched disturbance added into it becomes $\dot{x}_3 = 936.39x_2 - 41.19x_3 + 72.4593u + d(t)$ and $\dot{x}_1 = x_3$, where $d(t) \triangleq 9\,\text{deg}$ from $11 \leq t < 12$ s and $d(t) \triangleq 0\,\text{deg}$ from $12 \leq t < 14$ s and the disturbance matrix $B_f$ becomes $[0\ 0\ 1\ 0]^T$.

EGESO is used to estimate all four states and extended lumped disturbance. The sliding surface design for this system, considering all initial conditions of $\hat{x}(0)$ are zero, is obtained as

$$\begin{aligned} \overset{\star}{\sigma}_m &= \sigma - \sigma(0)e^{-\alpha_s t} \\ &= c_s(\hat{x} - \mathbf{r}) + \hat{x}_5 - (c_s\hat{x}(0) - c_s \cdot \mathbf{r} + \hat{x}_5)e^{-\alpha_s t} \end{aligned} \tag{43}$$

where $\hat{x}_5$ is an estimation of disturbance $d(t)$, $\hat{x} = [\hat{x}_1 \ \hat{x}_2 \ \hat{x}_3 \ \hat{x}_4]^T$, r is a set point for servo motor gear angle $\theta$ and gain matrix $c_s = [5 \ -24.3847 \ 1.0 \ -0.2]$. Equivalent control law $u_{eq}$ and disturbance caring term $u_n$ is obtained as

$$u_{eq} = -(c_s B)^{-1} \cdot (c_s \cdot A\hat{x} - \alpha_s c_s \mathrm{r} e^{-\alpha_s t} + \hat{\dot{x}}_5 e^{-\alpha_s t} + k_s \cdot (\overset{\star}{\hat{\sigma}}_m) + \hat{x}_5) \tag{44}$$

$$u_n = -(c_s B)^{-1} \cdot (c_s \cdot B_f \hat{x}_5 - \alpha_s \hat{x}_5 e^{-\alpha_s t}) \tag{45}$$

where the sliding surface coefficient $k_s = 30$, observer gain obtained as $[51.81 \ -27.2945 \ -146.3584 \ -245.1937 \ 234.7045 \ 743.9036]$, $\alpha_s = 4.5$, step size of simulation = 0.001 s, solver used in simulation is ode4 (Runge–Kutta). The proposed controller response is compared with other methods, and set-point tracking of $\theta$ and arm end-point deflection angle ($\alpha$) are observed as shown in Figure 10. It is found that the proposed controller is not giving any overshoot as compared to the other two methods, so its effect on $\alpha$ seems to be very less.

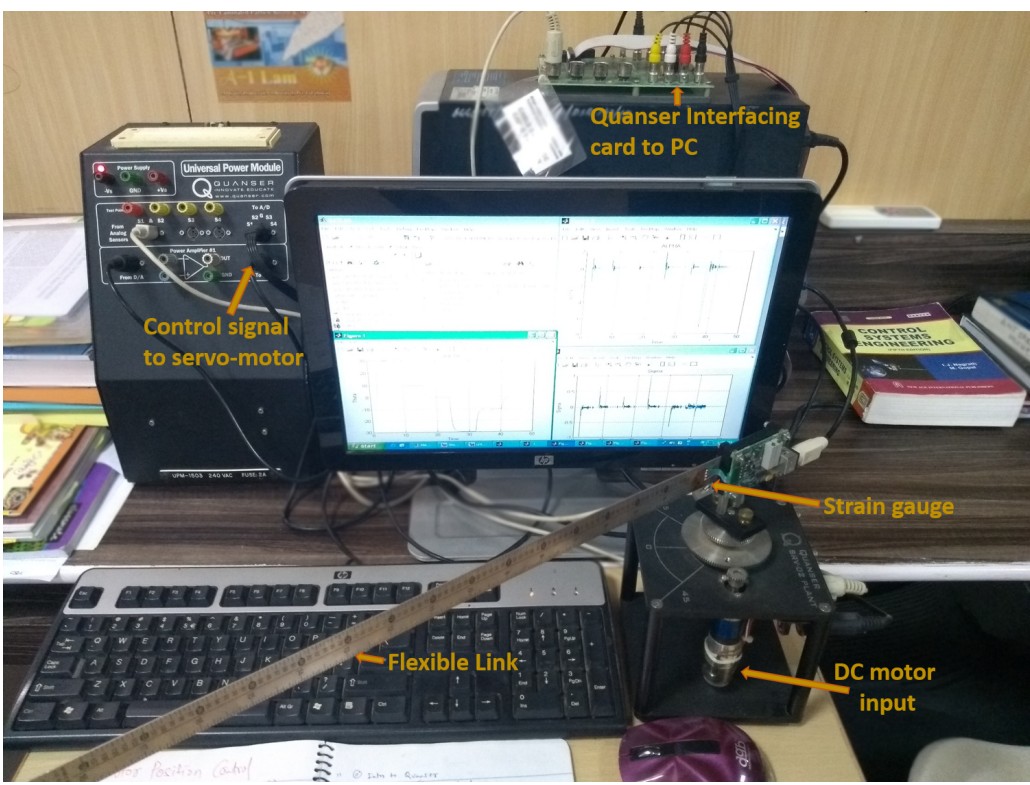

**Figure 9.** Flexible link experimental setup.

The proposed controller provides less control signal at the initial stage to provide smooth output $\theta$. It works better after external disturbances are added into the system at time $11 \le t < 12$ s. Figure 11 shows the control input for simulation results. The sliding surface response is shown in Figure 12 and it is observed that it is approaching zero.

The actual and estimated states are $\theta, \hat{\theta}, \alpha, \hat{\alpha}$ given in Figure 13. It is observed that GESO properly measures the estimated states, whereas the extended state estimates disturbance acts on the system immediately. The estimation of disturbance and error in disturbance estimation (error dynamics) are shown in Figure 14.

From the performance indices shown in Table 4, it is clarified that the proposed controller provides a minor error compared to other techniques in all cases. It is observed that the proposed scheme improves the control performance by reducing efforts of dc motor to 12.47%, 0.78%, and 65.80% as compared with GESO based method [13], EGESO based method [14], and Linear extended state observer-based sliding mode control (LESOSMC) [30] method respectively.

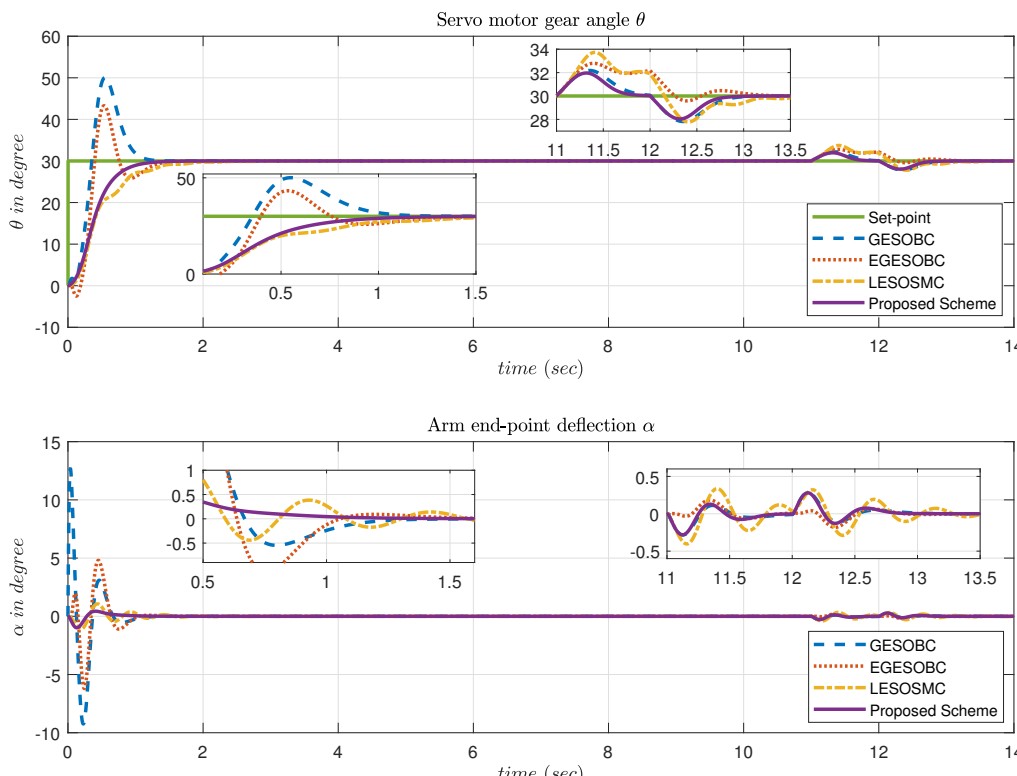

**Figure 10.** Simulation result of output tracking $\theta$ and arm end-point deflection $\alpha$.

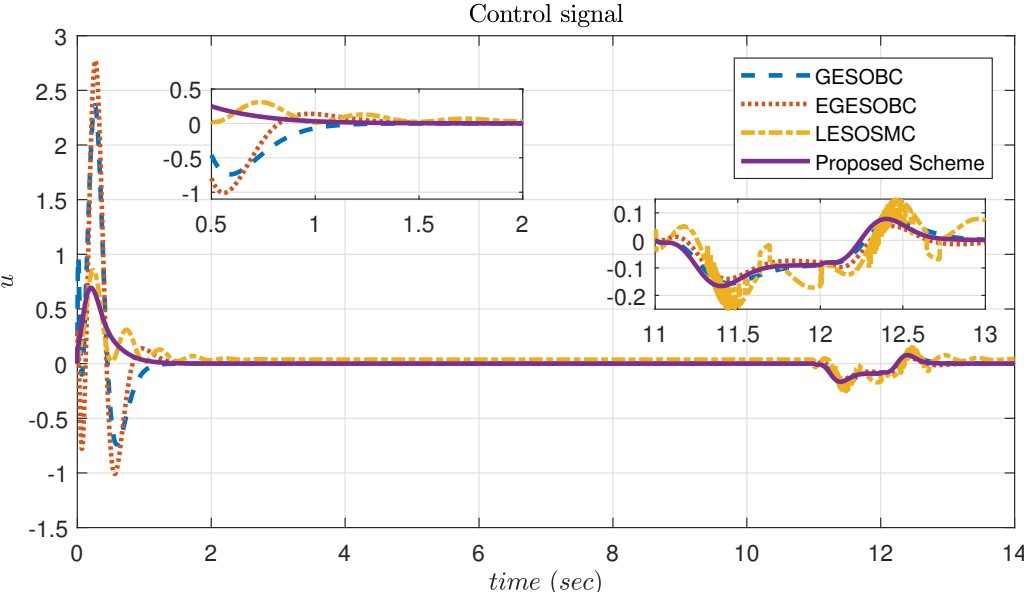

**Figure 11.** Simulation result of control signal for controlling $\theta$.

### 5.3.3. Experimentation Result for SLFM

The standard flexible link manipulator provided by [39] is fitted with a strain gauge sensor resulting in an analog signal proportional to the deflection of the arm end-point and servo motor sensor, which provides an angular position of arm as described in Figures 8 and 9. In this experimentation, only one sensor output is measured, which is a servomotor angular orientation ($\theta$), and remaining states ($\alpha, \dot{\theta}, \dot{\alpha}$) are measured by EGESO along with disturbances. MATLAB [41] software is used for simulation purpose considering ode4 (Runge–Kutta) solver for differential calculations with a sample time of 0.001 s.

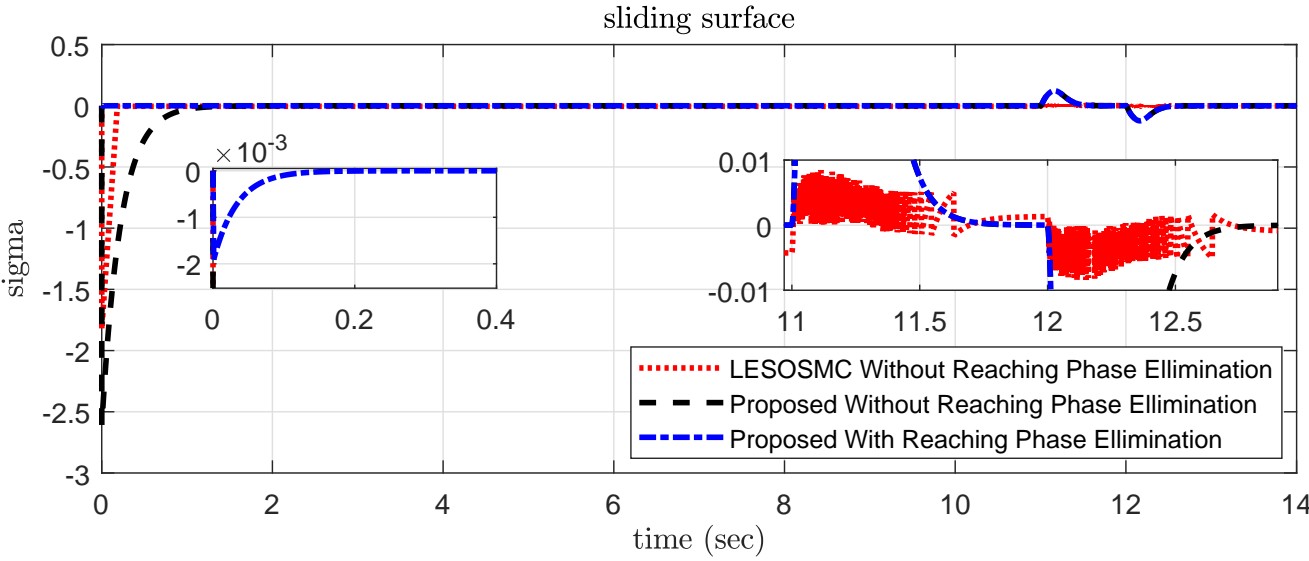

**Figure 12.** Simulation result of sliding surface $\sigma$.

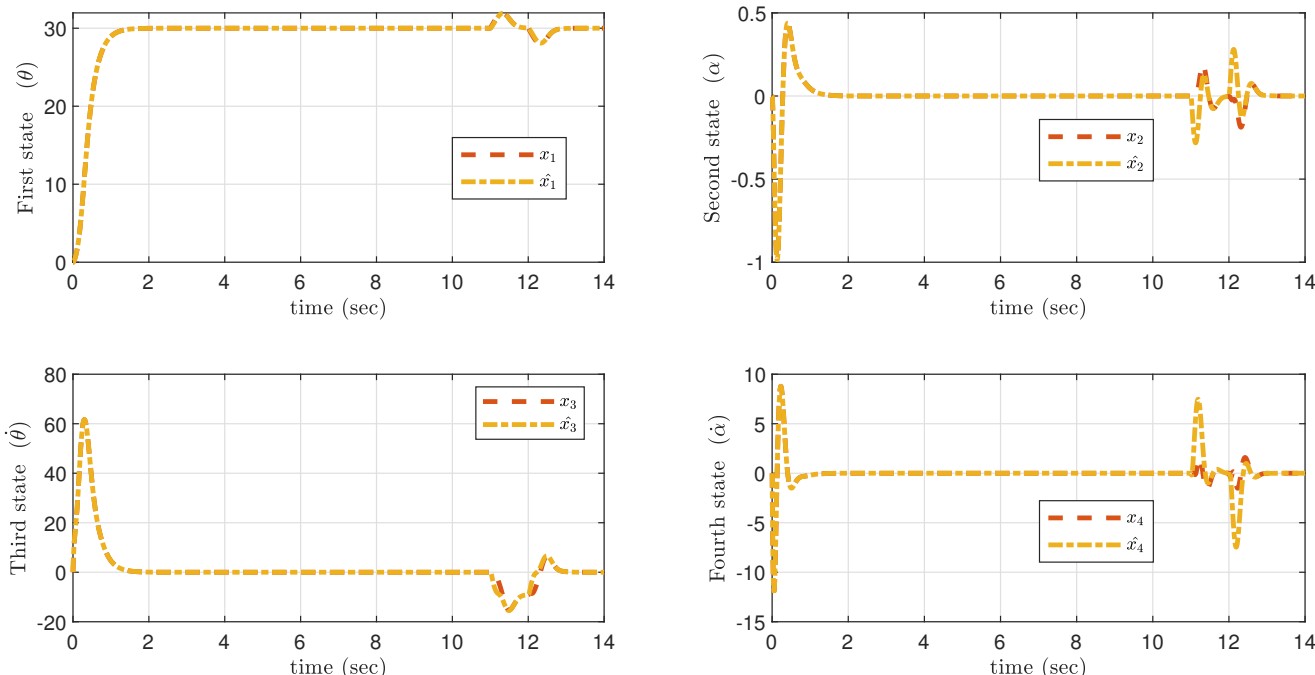

**Figure 13.** Simulation result of actual and estimated states after reach phase elimination.

In order to stress the effect of the difference in real-time control efforts, multilevel fixed set-points trajectory with disturbances is adopted. The servo motor gear angle $\theta$ for the selected trajectory is shown in Figure 15. On the zoomed portion, the proposed controller offers a smooth response without overshoot as compared to other techniques, which leads to less chattering in $\theta$. Furthermore, it was also considered to have the desired requirement of less arm end-point deflection ($\alpha$), and it is observed that the proposed scheme works effectively to satisfy the desired requirement in Figure 16.

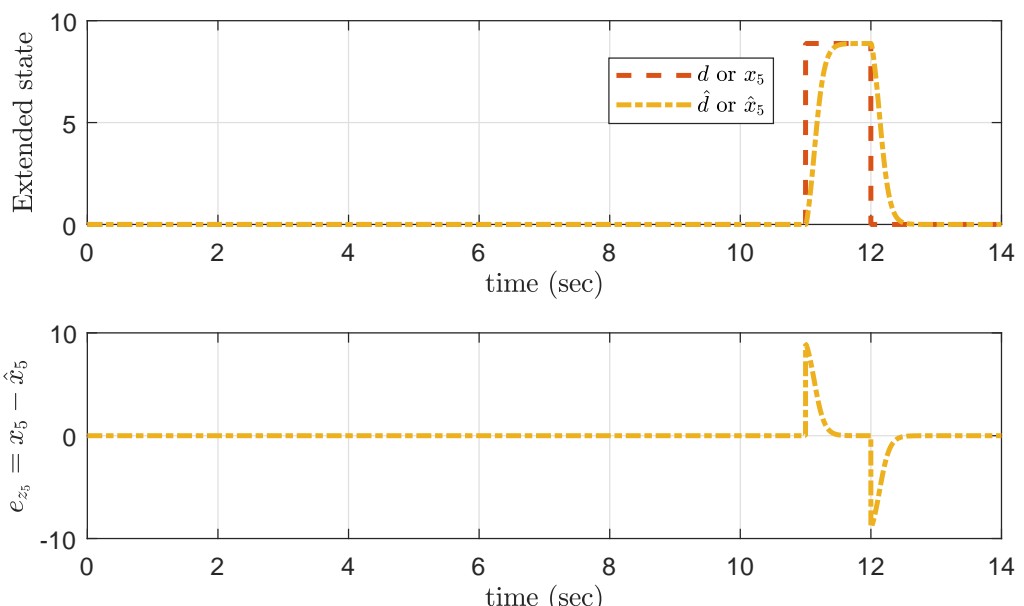

**Figure 14.** Simulation result of error dynamics.

**Table 4.** Comparison of performance indices for simulation results of SLFM.

| Controller | Set-Point ($r$) Radian Output ($y$) | Error Indices | | | | Control Efforts ($u$) in ITAE |
|---|---|---|---|---|---|---|
| | | IAE ($10^{-2}$) | ISE ($10^{-3}$) | ITAE ($10^{-2}$) | ITSE ($10^{-3}$) | |
| GESOBC | error = y − 0.523 rad | 29.50 | 88.51 | 54.59 | 37.66 | 1.885 |
| EGESOBC | error = y − 0.523 rad | 28.69 | 91.29 | 60.27 | 34.91 | 1.663 |
| LESOSMC | error = y − 0.523 rad | 34.22 | 91.02 | 89.72 | 47.24 | 4.825 |
| Proposed scheme | error = y − 0.523 rad | 25.11 | 78.97 | 42.06 | 23.29 | 1.650 |

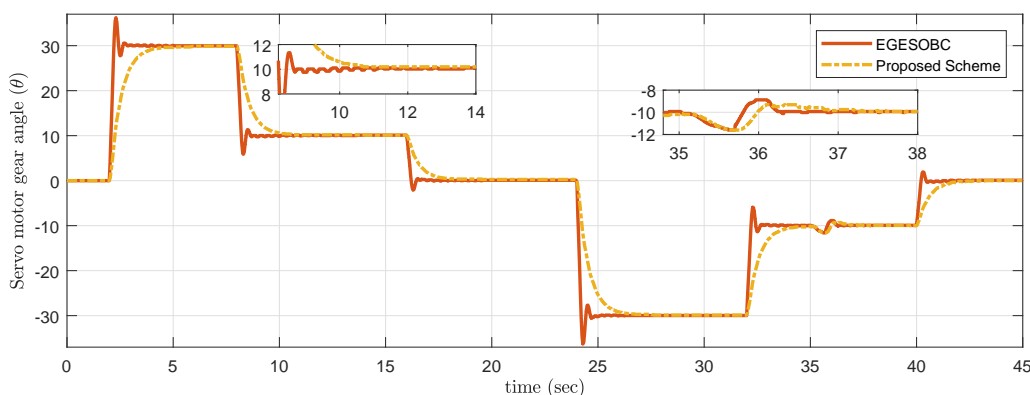

**Figure 15.** Experimental result of output tracking $\theta$.

Disturbance of small impulse added manually at time $t = 35$ s, and it is observed that the proposed controller is not giving much overshoot compared to other techniques.

The motor (actuator) response for controlling servo motor gear angle $\theta$ is shown in Figure 17. Compared to other techniques, the amplitude order of the proposed control signal is very low.

The sliding surface response is shown in Figure 18 and it is observed that chattering in sliding surface is of very low order, which is nearly approaching zero.

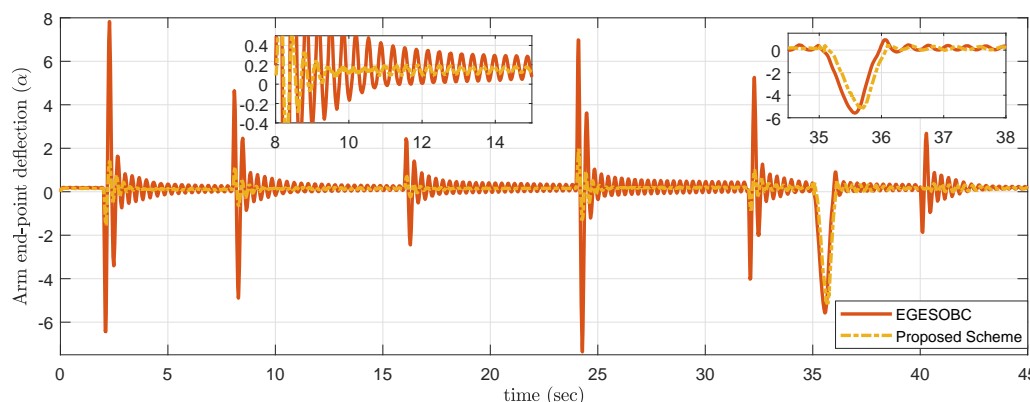

**Figure 16.** Experimental result of arm end-point deflection $\alpha$.

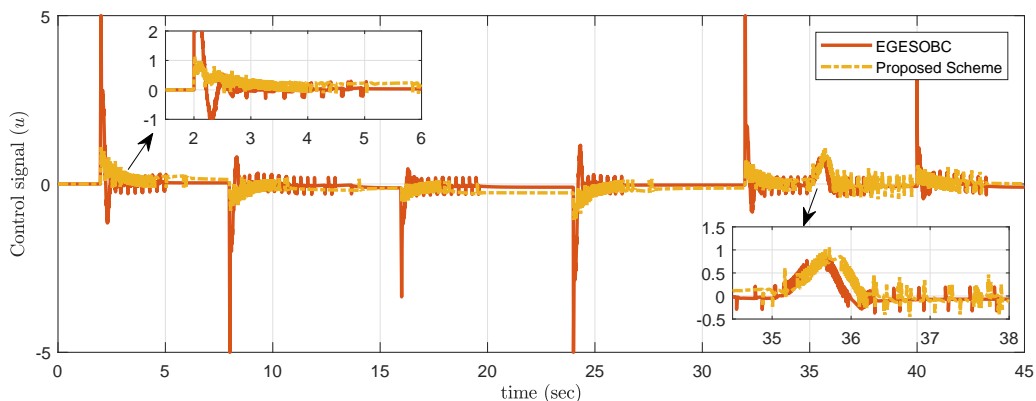

**Figure 17.** Experimental result of control signal for controlling $\theta$.

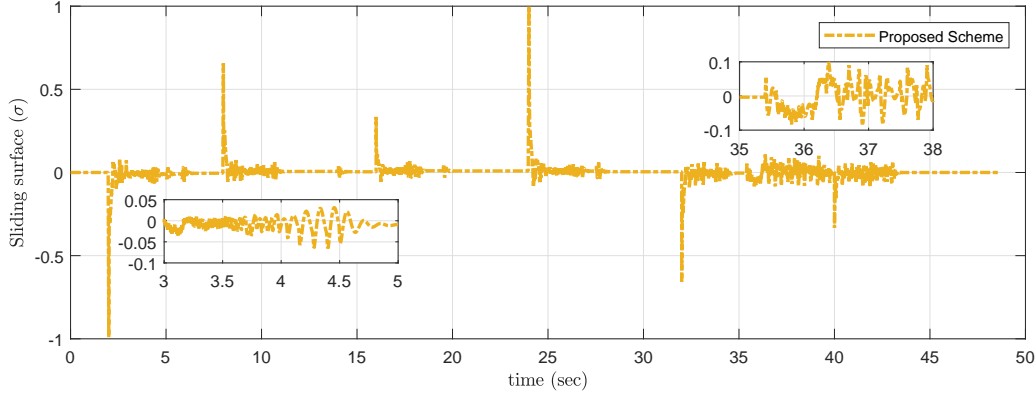

**Figure 18.** Experimental result of sliding surface $\sigma$.

## 6. Conclusions

In this paper, an EGESO based SMC is designed and applied to SLFM model. The proposed controller shows better transient performance with fewer overshoots than other existing GESOBC techniques, based on performance criteria. In the experimental case, an integral time absolute error (ITAE) has values 42.06, 54.59, 60.27 for the proposed control law, GESOBC, and EGESOBC, respectively. It is observed that ITAE for the proposed controller is 30.21% less than EGESOBC and 22.95% less than GESOBC; this shows the efficacy of the proposed controller. Due to the inclusion of reaching phase elimination in the proposed control method, the control efforts are significantly reduced. Finally, the proposed scheme could be extended to deal with high-frequency disturbances systems.

**Author Contributions:** Conceptualization, T.B.; methodology, T.B., H.F.H. and N.B.M.N.; software, T.B.; validation, T.B.; formal analysis, T.B. and R.H.C.; investigation, writing—original draft prepara-

tion, T.B.; writing—review and editing, T.B., H.F.H., N.B.M.N. and S.A.; supervision, H.F.H., N.B.M.N. and D.W.; project administration, H.F.H.; funding acquisition, H.F.H. All authors have read and agreed to the published version of the manuscript.

**Funding:** This research was funded by Yayasan Universiti Teknologi PETRONAS (YUTP) grant (number: 015LC0-153).

**Institutional Review Board Statement:** Not applicable.

**Informed Consent Statement:** Not applicable.

**Data Availability Statement:** Not applicable.

**Acknowledgments:** The authors would like to thank Universiti Teknologi PETRONAS (UTP) Malaysia, Shri Guru Gobind Singhji Institute of Engineering and Technology, Nanded, India for their support.

**Conflicts of Interest:** The authors declare no conflict of interest.

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
