# Peer review of "Sliding Mode Controller with Generalized Extended State Observer for Single Link Flexible Manipulator"

_applsci, doi:10.3390/app12063079_

Round 1

Reviewer 1 Report

The paper presents a generalized extended state observer (GESO)-based sliding mode controller (SMC) for a class of systems, and apply it to a single link flexible manipulator. The paper is very poorly structured and written, and the examples used to demonstrate the findings are problematic.

  1. The second and third sentences of the Abstract are repetitive.
  2. The nonlinearities and uncertainties in the system are both lumped under a single term d, which is a conservative approach, as this not only requires the nonlinearities to be bounded, but also increases the total bound on d, which restricts the feasibility of the scheme.
  3. The structure of Section 2 is very messy, and it is very unclear which portion is _only_ applicable to [13] or [14]. Furthermore, it is unclear if Remarks 1--3 are only applicable to [13], [14], or to the work presented in this paper. The notation used in this section is also unclear: what are P and m_{u_m}^{tot} after (6)? Also, the meaning of Remark 2 is not clear.
  4. In points 5 and 6 of subsection 3.1, `matrix (A,B,C)' should be `matrix triple (A,B,C)'.
  5. The Assumptions 1--3 should be motivated for their practicality and necessity. Furthermore, is the observability of (\bar{A},C) sufficient for the observability of (A_g,C_g)? This is important as it underlies the design later.
  6. Statement 1 should be renamed as Proposition/Theorem 1.
  7. What is σ_m at the start of subsection 3.3? Please define terms as they are used.
  8. Subsection 3.3 also appears to present results only for a specific system (12), and not general systems in the form of (1). Does this mean that the content in subsection 3.3 are only applicable to this system, or are they general results? Please also present results in a proposition/theorem-proof environment to make your arguments simpler to follow.
  9. It is unclear how the controller is a sliding-mode controller. SMCs typically rely on a discontinuous switching term, which is missing from the control inputs used in the scheme.
  10. The result in subsection 3.5 is not proven; reproduce the necessary results from the referred citation as is necessary to show (24) results in reaching phase elimination. Furthermore, use the symbol ∞ instead of inf (which can also mean infinimum).
  11. The layout of Section 4 should also be put into a proposition/theorem-proof structure.
  12. Section 5 should be incorporated into subsection 6.4, together with the experimental example. Notably, model (33) is completely linear, which does not sufficiently demonstrate the contribution of the presented work.
  13. The numerical examples in subsections 6.1 and 6.3 are problematic: how is e^{x_1} proven to be bounded (and the bound μ found). The matrix B_f for subsection 6.3 is also dimensionally incompatible with there being two disturbances d_1 and d_2.
  14. The experimental example assumes that the first channel of (33) is affected by disturbances - this is physically impossible, as _by definition_, d(θ)/dt = \dot{θ}.

Author Response

We would like to express our sincere thanks to the editor and reviewers for their valuable comments and suggestions. We have substantially revised the paper by carefully addressing all reviewer’s comments. A list of point-by-point statements of our response is given below and all main revisions are highlighted.
Many thanks for your consideration. The response to your comments is highlighted in red color in a revised manuscript.

Reviewer 2 Report

Authors present a good paper about sliding mode controller with generalized extended state observer for single link flexible manipulator.

I didn't detect any problems.

Author Response

We would like to express our sincere thanks to the editor and reviewers for their valuable comments and suggestions. We have substantially revised the paper by carefully addressing all reviewer’s comments. We have also completely revised the grammatical and other spelling errors and fully tried to clarify all sentences.

Reviewer 3 Report

This manuscript presents a composite controller based on a sliding-mode controller and a generalized ESO for a class of non-integral chain systems. Relevant simulation and experimental results are also presented. My comments are listed as follows:

  1. The name of the software used in the simulation tests must be given.
  2. The sampling period used in simulation and experimental tests must be given.
  3. All parameters selected for the tested controllers in simulation and experimental tests must be presented in Tables. Moreover, in order to show the fairness of the comparison, the authors should discuss how to select parameters for all tested controllers.

Author Response

We would like to express our sincere thanks to the editor and reviewers for their valuable comments and suggestions. We have substantially revised the paper by carefully addressing all reviewer’s comments. A list of point-by-point statements of our response is given below and all main revisions are highlighted.
Many thanks for your consideration. The response to your comments is highlighted in blue color in a revised manuscript.

Round 2

Reviewer 1 Report

While some of my comments have been addressed, many of my comments still have not - some remarks were even ignored outright.

  1. Regarding my previous comment 2: this is still a conservative approach, and employed only by few other works (e.g., [13], which the authors are extending their work off of). A remark motivating the assumption would need to be given, especially on its necessity and its practical significance.
  2. Regarding my previous comment 3: the structure of Section 2 is still unclear: it is not obvious which portions and assumptions apply to only [13] or [14], or the rest of the paper.
  3. Regarding my previous comment 5: the assumptions are still not motivated. The statement on the observability of (\bar{A},C) is still only necessary instead of sufficient as claimed by the authors.
  4. Regarding my previous comment 8: system (12) is one possible form of system (1). The analysis in section 3.3 only applies to this particular form; no proof extends to the case where e^{x_1} is not present, or if x_2 is pre-multiplied by some other factor. In fact, the example in subsection 5.3 does not fit this form (12).
  5. Regarding my previous comment 9: from what I understood of the authors' explanation, not using the signum/trapezoidal switching function results in less chattering. My point however is that without this switching function (or approximation thereof), the controller is not a sliding mode controller, and should be referred to by some other label.
  6. Regarding my previous comment 10: the proof for subsection 3.5 is still not clear; how is g(x,0) designed to satisfy this? If there is no theoretical proof for how this is achieved, then numerical simulations are just coincidences.
  7. Regarding my previous comment 13: the remarks that the disturbances being bounded need to be included in the paper itself. Furthermore, how can it still be proven to be bounded if this paper applies a different controller structure than in [13]?

Reviewer 3 Report

I have no further comments.